# Microglial Responses to Stress-Induced Depression: Causes and Consequences

**DOI:** 10.3390/cells12111521

**Published:** 2023-05-31

**Authors:** Ruqayya Afridi, Kyoungho Suk

**Affiliations:** 1Department of Pharmacology, School of Medicine, Kyungpook National University, Daegu 41944, Republic of Korea; 2Brain Korea 21 four KNU Convergence Educational Program of Biomedical Sciences for Creative Future Talents, Kyungpook National University, Daegu 41940, Republic of Korea; 3Brain Science and Engineering Institute, Kyungpook National University, Daegu 41944, Republic of Korea

**Keywords:** chronic stress, depression, microglia, neuroinflammation, proinflammatory cytokines

## Abstract

Chronic stress is a major risk factor for various psychiatric diseases, including depression; it triggers various cellular and structural changes, resulting in the alteration of neurocircuitry and subsequent development of depression. Accumulating evidence suggests that microglial cells orchestrate stress-induced depression. Preclinical studies of stress-induced depression revealed microglial inflammatory activation in regions of the brain that regulate mood. Although studies have identified several molecules that trigger inflammatory responses in microglia, the pathways that regulate stress-induced microglial activation remain unclear. Understanding the exact triggers that induce microglial inflammatory activation can help find therapeutic targets in order to treat depression. In the current review, we summarize the recent literature on possible sources of microglial inflammatory activation in animal models of chronic stress-induced depression. In addition, we describe how microglial inflammatory signaling affects neuronal health and causes depressive-like behavior in animal models. Finally, we propose ways to target the microglial inflammatory cascade to treat depressive disorders.

## 1. Introduction

Major depressive disorder (MDD) is one of the most heterogeneous neuropsychiatric disorders that affects approximately 280 million people worldwide [1]. The disease is characterized by several core behavioral symptoms, including anhedonia, low self-esteem, sleep disturbances, and suicidal ideation [2]. Structural and functional abnormalities have been identified in the depressed brain, including decreased volumes of the prefrontal cortex (PFC) and hippocampus [3]. Various mechanisms have been proposed for decades to regulate depressive behaviors, most of which focused on restoring neuronal health and activity [4,5,6]. The monoamine hypothesis garnered the most attention; therefore, most antidepressant drugs aimed to replenish monoamine levels in the brain. However, the limited effectiveness of classical antidepressants in patients suggests the involvement of more intricate and multifaceted pathologies in depression.

Chronic stress is a major risk factor for MDD and has been associated with increased hypothalamic-pituitary-adrenal (HPA) axis activity and inflammatory activation of immune cells [7]. Increased levels of proinflammatory cytokines in the serum of depressed patients have inspired neuroscientists to investigate the possible role of neuroinflammation in depressive behavior [8]. This ushered in a new era in the neuropsychiatric field, leading to identifying brain immune cells, particularly microglia, as one of the prominent regulators of inflammation in the depressed brain. Microglia are the second major type of glial cells and primary immune cells that guard the brain parenchyma; these highly receptive cells respond to any changes in the brain microenvironment and adopt various structural and functional phenotypes in a context-dependent manner. In addition to their immune functions, microglia also regulate neuronal functions. For example, microglia aid in forming neural circuits in the developing brain through synaptic pruning and stripping, secretion of neurotrophic factors, and phagocytosis of dying neurons [9]. Microglial pruning of synaptic elements has also been implicated in cognition [10]. In the adult brain, microglia regulate dendritic spine formation via BDNF signaling, which plays a role in learning and memory [10]. Other brain cells, such as neurons and astrocytes, engage with microglia to maintain brain homeostasis [11].

Brain samples of patients with MDD have shown that chronic exposure to stressful stimuli affects microglial activation states [12]. Significant inflammatory activation of microglia was found in stress-responsive brain regions of MDD patients, including the PFC, nucleus accumbens, amygdala, and hippocampus [13,14,15]. Similar findings were also found in animal models of chronic stress-induced depression, where microglial inflammatory activation was positively correlated with depressive-like behaviors [16]. Proinflammatory cytokines released by inflammatory microglia can bind to their cognate neuronal receptors in stress neurocircuitry and regulate behavior [17,18,19]. Moreover, these cytokines can decrease the levels of neurotrophic factors, negatively impacting neuronal health. What exactly triggers microglial activation in stress-related depression remains unclear, though various pathways likely drive microglial immune activation, which has been reviewed previously [20,21,22]. In the current review, we discuss recent findings on the role of microglia in the pathology of stress-induced depression. We also discuss possible triggers of microglial responses in stress-induced depression. Finally, we identify several potential therapeutic targets that can be exploited to subdue microglial inflammatory activation in stress-induced depression. 

## 2. Molecular Mechanisms Underlying Microglial Activation in Depression

Microglia are brain resident immune cells, providing first-line defense from tissue damage and infection. Microglia are yolk-sac-derived, self-renewing myeloid cells and constitute around 10% of brain cells [23]. Microglia are in a constant surveillance state, playing a crucial role in brain homeostasis [24]. Microglia are endowed with a wide array of gene clusters, together called sensome, that can sense and respond to a variety of signals [25]. The presence of dynamic sensome enables microglia to transit to various functional phenotypes in a stimulus-dependent manner [25]. In neuroinflammatory conditions, microglia undergo extensive morphological and functional changes primarily to neutralize ongoing inflammatory processes [26]. The immune functions of microglia are proven to be beneficial in acute neuroinflammatory settings as well as neurodegenerative diseases [27]. However, increased inflammatory microglia activation for chronic periods results in many deleterious outcomes and exacerbates ongoing neuropathology.

Recent evidence strongly advocates inflammatory microglia activation in MDD and has been associated with negative affective states [28]. Despite the growing body of research on the involvement of microglia in depressive disorder, there is still a lack of understanding regarding the mechanisms by which inflammatory signaling is triggered in microglia. Various molecular pathways have been identified in microglia that lead to proinflammatory cytokine surges in various brain regions in animal models of chronic stress-induced depression [29,30].

The crucial role of microglial immune receptors has been identified in activating inflammatory signaling in microglia, including toll-like receptors (TLR), receptors for advanced glycation end products (RAGE), and purinergic receptors in animal models of stress-induced depression-like pathology [29,30]. These pattern recognition receptors can be activated by diverse classes of closely related damage-associated molecular patterns (DAMPs), which can lead to the induction of various signaling pathways, increasing proinflammatory cytokine release by microglia. In various animal models of stress-induced depression, the targeting of TLR4 receptors has decreased depressive-like behaviors [31,32,33]. Numerous inflammatory pathways, including glycogen synthase kinase-3 (GSK3), nuclear factor kappa-light-chain-enhanced of activated B cells (NF-κB), and nucleotide-binding domain, leucine-rich repeat, and pyrin domain-containing protein 3 (NLRP3) can be activated following pattern recognition receptors stimulation in microglia [32,33]. In various animal models of stress-induced depression, activation of NLRP3 signaling positively correlates with depressive-like behavior [30,34]. Activation of the NLRP3 pathway leads to the assembly of a multiprotein complex called the NLRP3 inflammasome, which activates caspase-1 and subsequently promotes the maturation and release of proinflammatory cytokines, such as interleukin-1β (IL-1β) and IL-18 [35]. Activation of these pathways has been reported in various animal models of stress-induced depression, including chronic social defeat stress (CSDS), chronic unpredictable mild stress (CUMS), and chronic restraint stress [33,36]. 

Recent studies indicate the role of metabolic pathways in regulating microglial functional phenotypes [37]. The switch between oxidative phosphorylation and glycolysis determines the inflammatory status of microglia in various neuroinflammatory and neurodegenerative diseases [38]. Activating the TLR4 receptor by inflammatory stimulus increases glycolytic metabolism in microglia, leading to increased proinflammatory cytokine release [39]. In addition, increased expression of translocator protein 18 kDa (TSPO), a well-known marker of inflammatory activation of microglia, is also documented in both preclinical and clinical studies of depression [40,41,42]. Interestingly, TSPO is also known to alter microglial bioenergetic pathways and increases the synthesis and release of reactive oxygen species (ROS) [43]. 

## 3. Possible Triggers of Microglial Inflammatory Activation in Stress-Induced Depression

Tissue damage or infection can induce microglial inflammatory activation by releasing proinflammatory mediators [44]. In contrast, the exact triggers of microglial inflammatory activation under sterile inflammatory conditions such as chronic psychosocial stress are unknown. Chronic psychosocial stress causes cellular and structural changes in the brain, resulting in altered neurocircuitry and depressive behavior [41,45,46]. In the following sections, we discuss inflammatory factors that trigger the activation of microglial cells in response to psychosocial stress in animal models (Figure 1). Animal models used in preclinical studies of depression often use stressors to induce depression-like pathology [47]. While some of these models may not accurately reflect the actual pathophysiology of human depression, they consistently exhibit features such as hyperactive HPA axis, impaired neuroplasticity and neurogenesis, and altered neurotransmitters that can be related to human depression [28,48,49,50]. These models have greatly contributed to our understanding of depression, particularly in revealing the role of neuroinflammation in its pathophysiology [47].

### 3.1. Hyperactivity of HPA Axis

Neuroendocrine responses to psychosocial stressors are an important compensatory mechanism [51]. The classical “fight-or-flight ” response leads to hyperactivity of the HPA axis, increasing circulating glucocorticoids and catecholamines that return to baseline levels after the threat wanes [52]. In chronic psychosocial stress, however, persistently increased HPA activity exerts deleterious effects on the brain [53]. Chronic stress can lead to maladaptive changes in the HPA axis, which can contribute to the development of depression [52]. In vivo models have shown that the upregulation of HPA activity varies according to the type of stressor, which also reflects variable levels of glucocorticoids in a subset of patients with MDD [53]. 

The brain responds to stress by identifying potential threats and triggering corresponding physiological and behavioral reactions that can either be beneficial or harmful [51]. The brain is the main target of glucocorticoid actions, which are elevated after exposure to stressful stimuli [51]. All major cell types of the brain, including neurons, astrocytes, and microglia, express glucocorticoid receptors [54,55]. These receptors are expressed in distant limbic–midbrain and cortical brain regions, including the hippocampus, amygdala, and prefrontal anterior cingulate cortex, suggesting the role of glucocorticoids in stress-related mood disorders [56]. Importantly, distinct hippocampus regions show varied sensitivities to glucocorticoid activity [50]. The hippocampus plays diverse roles in memory and behavior due to functional segregation along its longitudinal axis. The dorsal hippocampus primarily contributes to spatial learning and memory, whereas the ventral hippocampus mainly regulates anxiety, which is influenced by stress. Due to its direct connection to the hypothalamus, the ventral hippocampus is more prone to the deleterious effects of glucocorticoids compared to the dorsal hippocampus [57].

Given the impact of the HPA axis on stress neurocircuitry, the increased HPA axis may drive the phenotypic transition of microglia in chronic stress-induced depression [49]. Indeed, recent literature has reported that increased HPA axis activity drives the primed state of microglia and induces the inflammatory phenotype in stress-sensitive brain regions [49,58,59]. Increased serum glucocorticoid levels were found in both preclinical and clinical studies of MDD [60]. Glucocorticoids also increased NLRP3 inflammasome signaling in the hippocampal region of mice subjected to chronic restraint stress [58]. Increased levels of high-mobility group box 1 (HMGB1) in limbic regions of the rat brain were reported in a model of inescapable tail shock, where subsequent administration of an antagonist blocked glucocorticoid signaling and attenuated the increase of HMGB1 levels [59]. In addition, increased inflammatory signaling in microglia was observed in a mouse model of corticosterone-induced depression [61,62]. Increased levels of proinflammatory cytokines in microglia were accompanied by depressive-like behavior in mice injected with corticosterone [63].

### 3.2. Peripheral Signals: Brain-Immune Axis

The brain is a distinct structure separated from the rest of the body by the blood-brain barrier (BBB). BBB acts as a selective barrier, regulating peripheral access to the brain parenchyma. The loss of BBB integrity has been documented in MDD pathophysiology, but the role of peripheral signals in microglial activation in vivo is debatable [64]. Immune dysfunction has been documented in patients with MDD as well as in preclinical models of depression [65,66,67]. Peripheral immune cells are the major sources of circulating proinflammatory cytokines that can induce inflammatory activation of microglia [68]. In addition, peripheral immune cells infiltrate the brain parenchyma in various animal models of depression [69]. Whether these peripheral immune cells trigger microglial activation in depression is unclear.

In a CSDS model, susceptible mice exhibited decreased expression of claudin-5, a tight junction protein in the BBB, which allows peripheral immune cells and proinflammatory cytokines to enter the brain [48]. Moreover, transcriptomic analysis of endothelial cells from susceptible mice revealed increased expression of genes associated with the proinflammatory tumor necrosis factor-α (TNF-α) and the NF-κB pathway [48]. The study also found decreased claudin-5 expression in post-mortem samples of patients with MDD [48]. Thus, a compromised BBB can allow proinflammatory signals from the periphery to act on microglia in a depressed brain.

Increased trafficking of monocytes to the perivascular space and parenchyma was also observed in the repeated social defeat model [69]. Using chimeric mice expressing the green fluorescent protein in lysozyme M (LysM)-positive myeloid cells, the study found an increased infiltration of monocytes in various brain regions of defeated mice. Interestingly, significant increases in IL-1β, chemokine (C-C motif) ligand 2 (CCL2), and microglial activation were also found in brain regions in which peripheral macrophages infiltrated [69]. Finally, the study found that crosstalk between chemokine receptor-2 (CCR2) and fractalkine receptor (CX3CR1) recruits macrophages to the brain parenchyma under stressful conditions [69]. 

Contrary to these findings, another group found that peripheral immune cells do not play a role in microglial inflammatory activation in acute and chronic social defeat stress models [70]. Here, chronic social defeat increased phagocytic microglial cells in the brain without recruiting peripheral immune signals, indicating that microglia are solely responsible for generating inflammation in the brain during chronic stress. Moreover, peripheral signals can attenuate pathways involved in monoamine synthesis. A recent study showed that lipopolysaccharide-binding protein (LBP) expression increased both peripherally and centrally in mice following exposure to stressful stimuli [71]. LBP expression also increased in microglial cells and inhibited enzymes involved in synthesizing monoamine neurotransmitters. These findings suggest a bidirectional communication between neuroendocrine stimuli and the immune system in the pathology of depression.

### 3.3. Neuronal Signals Shape Microglial Responses

Microglia and neurons work together by secreting diverse molecules to regulate brain homeostasis. Particularly, neuronal-derived soluble factors, including colony-stimulating factor 1 (CSF1), CX3CL1, and transforming growth factor-β (TGFβ), play crucial roles in regulating microglial immune functions [72,73]. Dysregulation of neuronal activity and neuronal atrophy following stress alters neuronal-derived factors that maintain microglial activity, leading to increased inflammatory signaling in microglial cells [74]. Mice exposed to the chronic unpredictable stress model displayed increased expression of CSF1 in the PFC as well as CSF1 receptor (CSF1R) in microglial cells in the same region [72]. Augmented CSF1 signaling in microglia increased phagocytosis of neuronal elements, which reduced dendritic spine density. Interestingly, the knockdown of neuronal CSF1 decreased microglial phagocytosis and attenuated behavioral deficits in stressed mice. Impaired CX3CL1-CX3CR1 signaling between neurons and microglia has also been shown in an animal model of chronic stress, causing inflammatory activation of microglial cells [75]. Microglial deletion of CX3CR1 prevented mice from developing depression-like behavior after stress exposure. Ultimately, CX3CR1 deficiency attenuated chronic stress-induced proinflammatory gene expression in microglia and prevented neuronal dysfunction [76].

### 3.4. Role of Damage-Associated Molecular Patterns (DAMPs) 

Studies have reported that damage-associated molecular patterns (DAMPs), including heat shock proteins, HMGB1, and S100 proteins, can initiate sterile neuroinflammatory processes in animal models of chronic psychosocial stress [77]. Microglia can recognize these DAMPs and transmit signals to intracellular NLRP3 inflammasomes through TLRs and RAGE [29,66]. These immune receptors have been shown to promote microglial inflammatory signaling in an animal model of chronic stress and depression.

In addition to increased mRNA levels of HMGB1 in hippocampal microglia, higher expression of RAGE and activation of NLRP3 inflammasomes were found in the CUMS model of depression [29]. The increased HMGB1-RAGE signaling in hippocampal microglia coincided with depressive-like behavior in mice exposed to chronic unpredictable stress. Increased expression of S100a8 and S100a9 was also found in the PFC of susceptible mice subjected to repeated social defeat stress [66]. Microglia-specific reduction of TLR2/4 expression by using a viral strategy, however, prevented mice from developing depressive-like behavior after repeated social defeat stress [66]. Increased HMGB1 expression was also observed in the hippocampal region in rat brains following inescapable tail shock. Increased HMGB1 expression positively correlated with heightened NLRP3 inflammasome signaling [78]. 

Chronic stress can trigger not only previously well-recognized DAMPs but also extracellular nucleosomes and histones. Increased histones and nucleosomes were found in the cerebrospinal fluid of the CUMS mice, which positively correlated with IL-1β levels in PFC [79]. Higher levels of nucleosomes promoted microglial inflammatory signaling in a C-type lectin receptor 2D (Clec2d)-dependent manner, increasing oxidative stress and IL-1β secretion [79]. Knockdown of Clec2d in PFC reduced microglial inflammatory activation and depressive-like behavior in CUMS mice. 

The studies discussed in this section show that various DAMPs are released in the brain following chronic exposure to psychosocial stressors, indicating that the sterile neuroinflammatory environment drives depressive-like behavior. However, the cellular sources of these DAMPs are still unclear. Given the multifactorial pathology of depression, neurons or glia may be major sources of DAMPs in a depressed brain. Indeed, increased HMGB1 mRNA has been found in neuronal and microglial cells of animals exposed to chronic stress [29,66], and microglia release HMGB1 in vivo under chronic stress, which can amplify inflammatory signaling in the stressed brain [78]; however, future research should confirm this. 

## 4. Role of Microglial Inflammatory Signaling in the Pathology of Stress-Induced Depression

Hypotrophy of various brain regions, including the PFC and hippocampus, has been observed in depressed patients, indicating altered structure and function of these brain regions [67,80]. Lower brain volumes in patients with MDD are partly due to altered synaptic connectivity and synaptic loss [45]. In a rodent model of stress-induced depression, microglial inflammatory activation positively correlated with neuronal damage [65]. Inflammatory microglia can damage neurons by releasing proinflammatory mediators, increasing phagocytosis of neuronal elements, and reducing the release of neurotrophic factors [46].

Microglia are an active source of proinflammatory mediators, which tend to regulate depressive-like behavior by changing neuronal excitability [81]. One mechanism by which activated microglia can induce depressive-like behavior in mice is to change neuronal excitability by releasing proinflammatory mediators [82]. In the CSDS model, the generation of microglial ROS positively correlated with depressive-like behavior in mice. The depletion of CSF1R reduced the overall ROS content in the brain and CSDS-induced behavioral deficits [48]. Furthermore, mice whose microglia exhibit increased inflammatory gene expression are more susceptible to depression, further linking microglia to CSDS-induced behavioral changes [82]. Increased levels of prostaglandin E_2_ (PGE_2_) and IL-6 were found in the serum and cerebrospinal fluid of depressed patients [83,84,85]. Microglial inflammatory signaling modulates stress neurocircuitry involved in mood regulation as well as stress-induced behavioral outcomes. A recent study demonstrated the crucial role of microglial inflammatory signaling in depression-like behavior by explicitly controlling microglial activity, where chemogenetic activation of microglia in the dorsal striatum of mice led to aversive behavior [81]. Mechanistically, increased inflammatory signaling in striatal microglia enhanced secretion of IL-6 and PGE_2_, reducing the excitability of mid-spiny neurons and causing aversive behavior. These findings highlight the critical role of microglial inflammatory signaling in depression [86].

Inflammatory activation of microglia also reduces dendritic spine density in brain regions such as the PFC and hippocampus. Impaired synaptic plasticity was found in preclinical and clinical investigations of stress-induced depression [32]. Microglia can disrupt synaptic plasticity and spine density through the release of factors that impair neurotrophic support [35]. Activated microglia can release inflammatory factors that interfere with the expression and signaling of BDNF, as well as increase phagocytosis of synaptic elements [22]. A recent study observed increased NLRP3 inflammasome and NF-κB signaling in hippocampal microglia of mice exposed to chronic mild stress (CMS) [35]. Activation of NLRP3 inflammasome and NF-κB signaling in microglial cells in the hippocampus of CMS mice increased the release of C1q, IL-1α, and TNF-α, which activate A1 astrocytes. Transcriptomic analysis of astrocyte reactivity revealed increased expression of neurotoxic astrocyte markers, including C3, Amigo2, Fkbp5, and Serping1 in the hippocampus. Consequently, increased C3^+^ astrocytes in the dentate gyrus (DG) augmented phagocytosis of neuronal elements, resulting in synaptic impairment, including altered synaptic density and lower levels of pre-and post-synaptic proteins. Impaired synaptic remodeling, increased synaptic loss, and reduced excitability of pyramidal neurons were observed in the PFC of defeated mice [46]. Interestingly, the increased CD68^+^ microglia were also found in the PFC of defeated mice. C3-expressing neurons induced the phagocytic microglia, suggesting that stress-induced cortical underconnectivity and behavioral abnormalities are associated with excessive synaptic pruning mediated by microglia.

Microglia are active regulators of adult neurogenesis [87,88]. The hippocampal DG and subventricular zone are the two brain regions where new neurons are generated throughout adulthood. Neural stem/progenitor cells (NSCs) undergo various processes to develop into new neurons that are integrated into neural networks. However, stressful experiences hinder adult neurogenesis; antidepressant therapies can positively affect neurogenesis [89]. The dysregulation of microglia and their inflammatory response can disrupt the microenvironment necessary for neurogenesis, leading to impaired neurogenesis and potentially contributing to depression pathology [88]. Inflammatory microglia impair neurogenesis in depression through several mechanisms, including the release of microRNA-enriched exosomes and direct engulfment of NSCs. Rats exposed to the CUMS paradigm displayed increased expression of miR-146a-5p in the DG region. Mechanistically, miR-146a-5p decreased the expression of important regulators of neurogenesis, including krüppel-like factor 4 and cyclin-dependent kinase-like 5, reducing neurogenesis and inducing depressive-like behavior after that [87]. Another recent study highlighted how microglial phagocytosis alters neurogenesis in the CSDS model [90]. Phosphatidylserine (PS)-expressing NSCs were found in the DG; NSCs express thermosensitive transient receptor potential vanilloid 4 (TRPV4), which was activated by CSDS. Subsequently, increased calcium levels exposed PS on NSCs engulfed by microglial cells, reducing neurogenesis. 

## 5. Microglia as a Potential Therapeutic Target for Treatment of Stress-Induced Depression

Inflammatory activation of microglia in various limbic brain regions is a hallmark of chronic psychosocial stress not only in rodents but also in humans [91,92]. Patients with MDD exhibit increased proinflammatory cytokines in cerebrospinal fluid, decreased neurogenesis, and impaired synaptic plasticity [91,92]. Inflammatory activation of microglia is strongly linked to neuronal deficits in MDD pathology; therefore, microglial inflammation is a potential therapeutic target for treating depression. Various strategies have been used effectively in in vivo models of depression to mitigate inflammatory activation of microglia (Table 1).

Inhibition of inflammatory signaling in microglia, such as NF-κB and NLRP3, has been shown to reduce depressive-like behavior in animal models of chronic stress [93,95]. NLRP3 inflammasome signaling is increasingly recognized to play a crucial role in the chronic psychosocial stress model [93]. NLRP3 inflammasome signaling in microglia increased levels of proinflammatory cytokine IL-1β in the PFC and hippocampus in mice exposed to depression-mimicking paradigms. In the CUMS model of depression, NLRP3 levels were increased in microglial cells of PFC [93]. The treatment of mice with an NLRP3-specific inhibitor MCC950 decreased inflammatory activation of microglia, as shown by lower levels of proinflammatory cytokines and less arborized microglia morphology in the PFC compared to vehicle-treated animals. Another group has also reported the beneficial effects of targeting the NLRP3 inflammasome in CUMS model [94]. NLRP3 signaling was increased in the hippocampus of CUMS mice, and treatment of mice with astragalin inhibited NLRP3 inflammasome signaling in microglial cells. Mechanistically, astragalin inhibited NF-kB, the upstream regulator of NLRP3 inflammasome, by increasing Sirtuin 1 (SIRT1) expression. SIRT1 is known to inhibit inflammatory signaling mainly by inhibiting NF-kB signaling [110]. The phosphodiesterase-4 (PDE4) inhibitor also exerted an antidepressant effect in CUMS mice by inhibiting HMGB1-induced activation of TLR4 and RAGE receptors. The PDE4 inhibition reduced NF-kB and NLRP3 signaling, decreasing proinflammatory cytokines in the hippocampus and PFC of CUMS mice [95]. 

Increasing neurotrophic signaling in microglia also rescues behavioral deficits in chronic psychosocial stress models [96]. Chronic mild stress decreases IL-4 signaling in the hippocampus together with increased inflammatory cytokines [96]. Recombinant adeno-associated virus (rAAV) vectors encoding murine IL-4 were administered in CMS mice to alternatively activate microglia and therefore reduce inflammatory signaling while increasing Arg1-positive microglia in the DG, which increased microglial-derived BDNF. Together, these factors increased neurogenesis and alleviated depressive-like behavior in CMS mice [96]. The treatment of CSDS mice with sulforaphane decreased microglial inflammatory activation and induced the neuroprotective phenotype of microglia. This effect of sulforaphane was associated with an increased BDNF transcription, which was regulated by Nrf2 and MeCP2 [97]. Recently, ketamine has also proved effective in reducing depression in treatment-resistant MDD patients. As shown in an animal model of chronic stress, ketamine ameliorated depression by increasing BDNF transcription in microglial cells [98]. 

Liver X receptors (LXRs), including LXRα and LXRβ, are nuclear receptors that decrease microglial inflammatory activation by repressing proinflammatory gene expression [106]. Both LXR-α and LXR-β are present in the brain. LXRs have significant involvement in brain function, influencing the expression of synaptic proteins, axonal growth, and neurogenesis [111,112]. Additionally, LXRs can inhibit the transcription of proinflammatory genes through interactions with NF-κB [113]. Surprisingly, in the CUMS model of depression, LXRβ expression was lower in the hippocampus and basolateral amygdala regions of the mouse brain [105,106]. Intraperitoneal administration of an LXRβ agonist in CUMS mice alleviated CUMS-induced behavioral despair and synaptic deficits [106]. Peroxisome proliferator-activated receptor gamma (PPAR-γ), a ligand-dependent transcription factor, is another pathway that can inhibit microglial inflammatory signaling whose expression decreases in microglial cells in various animal models of depression. Treatment of CMS mice with asperosaponin VI, a natural compound that increased the expression of PPAR-γ in hippocampal microglia, decreased the expression of proinflammatory cytokines; it also increased Arg1 expression, reducing depressive-like behavior in CMS mice [107].

Inhibition of microglial inflammatory signaling has also shown promising results in animal models of depression. Minocycline is another anti-inflammatory antibiotic that inhibits microglial inflammatory activation and exerts antidepressant effects on patients with MDD. A recent study reported that minocycline administration in CUMS mice reduced phagocytic microglia in the DG of the hippocampus, alleviating depressive-like behavior [101]. Minocycline can also inhibit the release of HMGB1 from microglia, attenuating cognitive decline and depressive-like symptoms in CUMS mice [102]. 

Other molecules have also been used in an animal model of depression to inhibit microglial inflammatory activation and reduce depressive-like behavior. IL-10 is an anti-inflammatory cytokine used to reduce the inflammatory activation of microglial cells [114]. In the animal stress model, the expression of IL-10 is reduced with a concomitant increase in depressive-like behavior in animals [108]. In the learned helplessness (LH) model, microglial inflammatory activation largely contributes to behavioral despair and impaired cognition [108]. These effects were found to be a consequence of reduced IL-10 production in LH susceptible mice. In contrast, the mice resilient to LH stress showed a comparatively increased population of IL-10-producing microglia. Interestingly, the complete elimination of microglial cells impaired cognition in resilient mice, which was rescued by IL-10 administration. On the other hand, the cognitive profile was improved in LH susceptible mice by IL-10 administration or complete microglial elimination. These interesting findings point toward the inherent complexity of microglial cells in response to stressful stimuli [108]. Dimethyl fumarate is another anti-inflammatory compound that acts as an antidepressant in the CUMS model of depression. The treatment of mice with dimethyl fumarate reduced proinflammatory cytokine expression as well as microglial immunoreactivity in the hippocampus of the CUMS mice [109]. 

## 6. Conclusions and Future Perspectives

Microglial inflammatory activation plays a key role in MDD pathology. Several preclinical and clinical studies of MDD have reported increased neuroinflammatory signaling in various brain regions that regulate mood and behavior. However, the exact triggers of microglial inflammatory signaling in MDD pathology remain unclear. Recent studies suggest the possible role of elevated glucocorticoids in inducing the inflammatory phenotype of microglia, but the underlying mechanisms are unknown. The brain-immune axis is another factor that can induce microglial inflammatory activation during psychosocial stress. 

Despite these ambiguities, inflammatory activation of microglia and neuronal deficits are consistently observed in animal models of stress-induced depressive-like behavior. Increased proinflammatory signaling in microglia hampers neuronal health through phagocytosis of synaptic elements, altered synaptic plasticity, and decreased neurogenesis. Therefore, reducing microglial inflammatory activation is a potential strategy for treating MDD. Further research is needed to elucidate the molecular mechanisms involved in microglia activation and their impact on neurons in stress-induced depression. Future studies should better identify specific therapeutic targets to treat MDD. 

## Figures and Tables

**Figure 1 cells-12-01521-f001:**
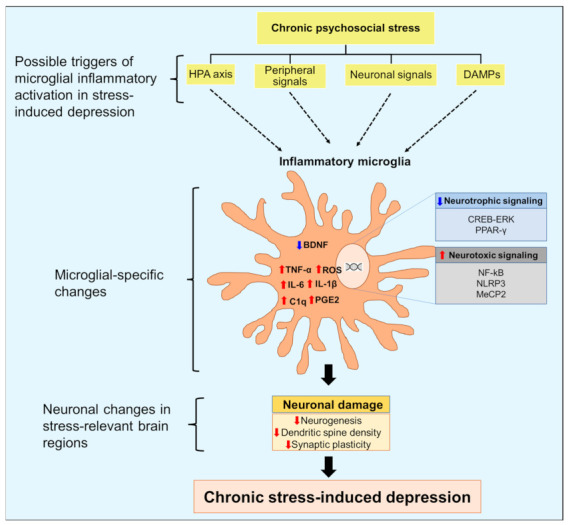
Triggers and role of microglial inflammatory activation in the pathogenesis of depression. Chronic psychosocial stress can increase hyperactivity of the hypothalamic-pituitary-adrenal axis, activation of peripheral immune cells, and release of damage-associated molecular patterns (DAMPs). Stress can also disturb communication between microglia and neurons that regulate microglial immune responses. Inflammatory signaling in microglia increases the expression of proinflammatory cytokines and the generation of reactive oxygen species (ROS). Inflammatory microglia also show decreased neurotrophic signaling, which hampers the release of brain-derived neurotrophic factor (BDNF) from microglia. These changes culminate in neuronal damage, including decreased neurogenesis, dendritic spine density, and impaired synaptic plasticity, leading to depression. ↓, decrease; ↑, increase; NF-κB, nuclear factor kappa-light-chain-enhancer of activated B cells; IL, interleukin; C1q, complement component 1q; TNF, tumor necrosis factor; MeCP2, methyl-CpG binding protein 2; CREB, cAMP response element binding protein; ERK, extracellular signal-regulated kinase; PPAR-γ, peroxisome proliferator-activated receptor gamma, NLRP3, nucleotide-binding domain, leucine-rich repeat, pyrin domain-containing protein 3.

**Table 1 cells-12-01521-t001:** Targeting microglial inflammatory activation in animal models of depression.

Putative Microglial Targets	Targeting Strategies	Animal Models	Brain Regions	Outcomes	References
↓ NLRP3 signaling	MCC950	CUMS (Mice)	PFC	↓ Depressive-like behavior ↓ Neuroinflammatory markers ↓ IL-1β	[93]
↓ NLRP3 signaling	Astragalin	CUMS (Mice)	Hippocampus	↓ Depressive-like behavior ↓ Neuroinflammatory markers ↓ IL-1β	[94]
↓ p38 MAPK signaling ↓ NF-κB signaling ↓ HMGB1/RAGE/TLR4 signaling	Roflupram	CUMS (Mice)	Hippocampus PFC	↓ depressive-like behavior ↓ proinflammatory cytokines	[95]
↑ BDNF signaling	Viral-mediated overexpression of IL-4	CMS (Mice)	Hippocampus	↑ Neurogenesis ↓ Depressive-like behavior ↓ Proinflammatory cytokines ↑ Arg-1 positive microglia	[96]
↑ BDNF by increasing Nrf2 signaling ↓ MeCP2 expression	Sulforaphane	CSDS (Mice)	PFC	↑ Resilience to stress ↑ Synaptic plasticity ↓ Proinflammatory cytokines	[97]
↑ ERK-NRBP1-CREB signaling ↑ microglial BDNF	(R)-Ketamine	CSDS (Mice)	PFC	↑ Dendritic spine density long-lasting antidepressant action	[98]
↓ NLRP3 signaling ↑ Autophagy	Ketamine	CRS (rats)	PFC Hippocampus	↑ Synaptic plasticity ↓ Depressive-like behavior	[99]
↓ CSF1 receptor expression ↓ CD11b ↓ (CR3)-C3 phagocytic pathway	Diazepam	CUS (Mice)	PFC	↑ Dendritic spine density long-lasting antidepressant action	[100]
↓ ERK 1/2 signaling ↓ Phagocytic microglia	Minocycline	CMS (Mice)	Hippocampus	↑ Neurogenesis ↓ Depressive-like behavior	[101]
↓ HMGB1 release		CUMS (Mice)		↑ Cognitive performance ↓ Depressive-like behavior	[102]
↓ Phagocytic microglia		CSDS (Mice)		↓ Proinflammatory cytokines ↓ Synaptic loss ↓ Behavioral despair	[103]
↓ Phagocytic and inflammatory microglia		CUMS (Mice)	PFC Hippocampus	↑ Kynurenic acid ↓ Behavioral despair	[104]
↑ LXR- β signaling ↓ NF-κB signaling ↓ NLRP3 signaling ↓ IL-1β ↓ Phagocytic microglia	TO90137	CUMS Corticosterone-induced depression	Basolateral amygdala	↓ Neuroinflammation ↓ Depressive-like behavior	[105]
↑ LXR- β signaling ↓ NF-κB signaling	GW3965	CUMS (Mice)	Hippocampus	↓ Inflammatory markers ↓ Synaptic impairment	[106]
↑ PPAR-γ signaling ↑ Neuroprotective microglia	Asperosaponin VI	CMS (Mice)	Hippocampus	↑ Microglial-neuronal interactions ↓ Synaptic deficits	[107]
Not discussed	murine recombinant IL-10	Learned helplessness (mice)	Hippocampus	↑ Dendritic spine density ↑ Cognitive performance	[108]
Not discussed	Dimethyl fumarate	CUMS (mice)	Hippocampus	↓ Neuroinflammatory markers ↓ Cognitive impairment	[109]

↓, decrease; ↑, increase; CUMS, chronic unpredictable mild stress; CSDS, chronic social defeat stress; CMS, chronic mild stress; CRS, chronic restraint stress; CUS; chronic unpredictable stress; PFC, prefrontal cortex; NLRP3, nucleotide-binding domain, leucine-rich repeat, pyrin domain-containing protein 3; MAPK, mitogen-activated protein kinases, NF-κB, nuclear factor kappa-light-chain-enhanced of activated B cells, IL-1β, interleukin-1β; HMGB1, high mobility group box 1; RAGE, receptors for advanced glycation end products; TLR, toll-like receptors; BDNF, brain-derived neurotrophic factor; PPAR-γ, peroxisome proliferator-activated receptor gamma, MeCP2, methyl-CpG binding protein 2; CREB, cAMP response element binding protein; ERK, extracellular signal-regulated kinase; LXR, Liver X receptors; CSF1, colony-stimulating factor 1.

## Data Availability

Not applicable.

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
