# Peer review of "Microglial Responses to Stress-Induced Depression: Causes and Consequences"

_cells, 2023, doi:10.3390/cells12111521_

Round 1

Reviewer 1 Report

The authors aim at summarizing the state-of-the-art literature about the role of microglia in depression and in stress-based animal models of depression. In particular, the authors focus on the contribute of the microglia in mediating the inflammatory processes that have been associated with depression and stress-induced maladaptive changes, trying to highlight the putative mechanisms involved. Moreover, an overview of potential therapeutic targets for the treatment of depression based on the modulation of microglia activation and function has been proposed.

The Review could offer an exhaustive description of the state-of-the-art literature on the topic yet providing useful insights into the mechanisms responsible for the microglia recruitment in neuroinflammation and how this process could be triggered by stress exposure.

In the literature, several other Reviews have already extensively addressed the topic (see for example: https://doi.org/10.1186/s12974-021-02309-6; https://doi.org/10.1111/ejn.14720, https://doi.org/10.1186/s12974-022-02492-0). Nonetheless, I think that the Review submitted by the Authors could still be valuable for the scientific community, but major changes need to be made.

First of all, the Authors always refer to “stress-induced depression”. Although stress has undoubtedly an important role in the etiopathogenesis of psychiatric disorders, including depression, the Authors should be very cautious in the use of the expression. I would recommend making a distinction between depression in humans and depressive-like behavior in stressed-based animal models throughout all sections of the Review. Beyond being more correct, it would help highlighting better whether the results described are from clinical or preclinical studies. This will help the readers not belonging to the field to have a clearer and unbiased picture.

Then, a brief but exhaustive introduction on microglia and its activity and functions in the brain is necessary to help the reader to better understand the other sections of the Review.

The Authors dedicated a section to the role of the HPA axis activation in the stress response, I would suggest referring at least once to the seminal works of Bruce McEwen in the field, giving the readers the opportunity to deepen the subject.

Finally, I found Table 1 very useful and simple to read. On the other hand, the relative paragraph is a bit confusing. Instead of just listing the same results that are already reported in the table, it would be better to discuss them thoroughly.

In general, since there are already Reviews that address this topic in a very detailed and systematic way, the Authors’ Review should really propose something new. Focusing on the molecular mechanisms that trigger microglia activation in depression is of relevance. For this reason, a better discussion of the results is necessary otherwise it looks like a list of the results of the several studies revised.

Minor points:

-line 92: correct “flight-or-fight” with “fight-or-flight”

-line 130: add references

-line 171: add references

-line 228: add references

-line 254: add references

-line 270: add references

Reviewer 2 Report

This manuscript reviewed microglial responses to depression caused by chronic stress from possible triggers to the potential therapeutic target. The writing idea throughout this manuscript is really clear. However, I noticed some parts need to be improved.

1. In "Abstract", I don't think "cellular and structural adaptations" in Line 11 is appropriate, as chronic stress already induced psychiatric diseases.

2. In "Introduction", I don't think "affect millions worldwide" in Line 27 is accurate, as I know approximately 280 million people in the world have depression.

3. I found in some paragraphs lack references, for example, Line 49-50, Line 70-71, Line 91-94, Line 109, Line 129-130, Line 187-189, Line 252-254.

4. I noticed that reference 54 and related content are not closely related to the current theme, although stress is a high risk factor for both anxiety and depression.

5. For "3. Role of microglial inflammatory signaling" part, Authors showed microglial inflammatory can damage neurongenesis, neuron excitability, synaptic density. But I found the structure of this part lacks logic.

6. I also noticed some language issue, like line 34-35, Line 126-127.

7. DAMPS in Line 78 should be "DAMPs".

Minor editing of English language required

Round 2

Reviewer 1 Report

I appreciate that the Authors implemented their manuscript with the suggested modifications. I think that now the Review has been significantly improved. 

I have some minor points:

-line 99: Change "Depressive-like symptoms" with "Depressive-like behaviors"

-Table 1: It could be useful to the readers if the order of the columns would be: Putative microglial target/Targeting strategies/Animal models/Brain regions/Outcomes/References
